# The Optimal Concentration of KH$_2$PO$_4$ Enhances Nutrient Uptake and Flower Production in Rose Plants via Enhanced Root Growth

**Qinghua Ma [1], Xinghong Wang [1,\*], Weijie Yuan [1], Hongliang Tang [2] and Mingbao Luan [3,\*]**

[1] Experimental Centre of Forestry in North China, Chinese Academy of Forestry, Beijing 102300, China; maqh2016@caf.ac.cn (Q.M.); yuanwj@caf.ac.cn (W.Y.)

[2] School of Life Science, Hebei University, Baoding 071002, China; thl_1980@163.com

[3] Institute of Bast Fiber Crops/Centre of Southern Economic Crops, Chinese Academy of Agricultural Sciences, Changsha 410205, China

\* Correspondence: hualinlinjun@caf.ac.cn (X.W.); luanmingbao@caas.cn (M.L.); Tel.: +86-010-69842397 (X.W.); +86-731-88998527 (M.L.); Fax: +86-010-69842397 (X.W.); +86-731-88998528 (M.L.)

**Abstract:** Monopotassium phosphate is a widely used phosphorus and potassium fertiliser for ornamental plants, but it is not known what concentration will result in optimal flower production, root growth and nutrient uptake of rose plants. We compared potted rose plants fertilised with six different concentrations (0.0 as a water-only control, 1.0, 2.0, 3.0, 4.0 and 5.0 g·L$^{-1}$) of an aqueous monopotassium phosphate solution as a combination of foliar and soil applications over two consecutive flowering cycles. Rose growth, flower production and nutrient accumulation responded differently to fertilisation with different concentrations of monopotassium phosphate. During the first flowering cycle, shoot and root dry weight, leaf chlorophyll content, flower diameter, total root length and surface area, and total fine root length significantly increased in response to increased monopotassium phosphate concentrations from 0.0 to 3.0 g·L$^{-1}$ but decreased in response to fertilisation with 4.0 or 5.0 g·L$^{-1}$ monopotassium phosphate. Similar trends were observed in shoot dry weight, leaf chlorophyll content, flower diameter and number, phosphorus and potassium accumulation during the second flowering cycle. According to quadratic equations derived from both flowering cycles, the optimal concentration of monopotassium phosphate, based on flower diameter and dry weight, as well as total phosphorus and potassium accumulation, was 2.6–3.0 g·L$^{-1}$. Furthermore, total root length was significantly correlated with flower diameter, flower dry weight, and total phosphorus and potassium accumulation ($p < 0.05$). These results indicated that fertilisation with optimal concentrations of monopotassium phosphate can increase rose growth, flower productivity and nutrient uptake through enhanced root growth.

**Keywords:** monopotassium phosphate; optimal concentration; flower production; root growth; P and K accumulation; *Rosa*



## 1. Introduction

Roses (*Rosa* spp.) have been cultivated by Chinese, western Asian and northern African civilisations for at least five thousand years [1]. The rose is the world's favourite flower because of its beautiful ornamental blooms available in a wide variety of colours [2]. Recently, there has been a pressing need to increase the quantity and quality of cut and potted flowers to meet the high market demand, especially in China [3]. Fertiliser plays an essential role in the production and productivity of rose plants. However, excessive fertilisation is widespread among small farmers in China [4]. Therefore, improved nutrient management has become an important issue in sustainable rose production.

Phosphorus (P) and potassium (K) are essential macronutrients for plant growth and development. P enables energy transfer throughout the plant for root development and flowering. K is essential for photosynthesis and many metabolic processes required for

growth and flower and fruit development [5]. P deficiency causes dull foliage, falling leaves, weak flower stems, and prevents buds from opening readily [6]. Furthermore, when K uptake falls below required levels, foliar potassium is mobilized to the fruit, reducing plant growth and fruit set and quality [7]. Adequate P and K nutrition have been associated with increased flowering, fruit size and fruit set [8,9]. Recently, monopotassium phosphate (MKP, $KH_2PO_4$) has been widely used as a P and K source in horticultural crop production [10].

Plants can absorb nutrients through either their roots or foliage. Foliar nutrient absorption serves as a route for remedial action to correct nutrient deficiencies [11]. Under various nutrient deficiencies, foliar application of nutrients enables instant transportation of nutrients to various plant parts from the leaf tissue [12]. Some studies reported that foliar application of MKP at critical periods (e.g., during reproductive development) resulted in greater flower diameter, yield and quality compared to controls [8,9]. In contrast, other studies reported that foliar fertilisation alone cannot fulfill the nutritional requirements of crops but can only supplement soil fertilisation [13]. An adequate supply of available macronutrients in the soil is a prerequisite for the optimal growth and reproductive success of plants. Foliar K application combined with basal soil K application led to higher yields than foliar K application alone [7]. Furthermore, because foliar fertilisation can only supply a limited amount of macronutrients to plants, foliar fertilisation should be combined with soil-based fertilisers to increase yields [11].

Root growth and root system establishment play a critical role in the capture and acquisition of resources such as nutrients and water and thus have a strong influence on plant growth, nutrient uptake efficiency and production quality [5]. Root system development is highly plastic in response to environmental conditions [14]. Lateral root proliferation is strongly related to the enhanced uptake of nutrients, especially P [15]. Hence, the manipulation of root morphology may improve plant growth and flower production.

Mineral status influences vegetative and reproductive growth, including flowering and fruit set, both directly and indirectly. Most studies reported a positive response in plant productivity to MKP fertilisation [16,17]. However, there was a marked difference in the reported concentration of MKP used for fertilisation. In some cases, there was a more than ten-fold difference in MKP concentration between studies [9,17,18]. Furthermore, little is known about the effects of MKP concentration on the root growth and flower production of rose plants. In the present study, we hypothesized that MKP concentration would influence rose flower production and nutrient accumulation by modifying root growth.

The aims of the present study were (1) to characterize rose plant growth, root morphology and P and K accumulation following fertilisation with different concentrations of MKP, (2) to determine flower production in roses fertilised with different concentrations of MKP, and (3) to analyse the correlation between flower production, P and K availability, root growth and MKP concentrations.

## 2. Materials and Methods

### 2.1. Plants, Soil Preparation and Experiment Design

The experiment was conducted at the Experimental Centre of Forestry in North China, Chinese Academy of Forestry, Beijing (39.55° N, 116.06° E). We transplanted 48 two-year-old *R. multiflora* plants with a basal diameter of 2.2–2.5 cm and a length of 45–50 cm from the field into pots to be used as rootstock. Two dormant buds of *R.* 'Pink Fan' were grafted onto each rootstock using the T-budding method and cultivated with conventional methods for a year before the present study.

The pots were 26 cm in upper diameter, 22 cm in bottom diameter and 24 cm in height. All pots were filled with a 10 kg mix of garden soil and sand (at a weight ratio of 75:25). The initial soil nutrient properties were: pH = 7.75 (1:2.5, soil:water), Olsen-P = 10.9 mg·kg$^{-1}$, available nitrogen (N) = 75.2 mg·kg$^{-1}$, organic carbon = 8.6 g·kg$^{-1}$, total N = 0.44 g·kg$^{-1}$, and exchangeable K = 80 mg·kg$^{-1}$. To ensure that the supply of other nutrients was adequate for plant growth, the soil was supplemented with the following basal nutrients

at the indicated rates: Ca (as 100 mg $CaCl_2$ per 1 kg soil), K (as 150 mg $K_2SO_4$ per 1 kg soil), Mg (as 50 mg $MgSO_4$ per 1 kg soil), Zn (as 6.0 mg $ZnSO_4$ per 1 kg soil), Cu (as 1.0 mg $CuSO_4$ per 1 kg soil), Fe (as 2.0 mg EDTA-Fe per 1 kg soil), Mn (as 2.0 mg $MnSO_4$ per 1 kg soil), B (as 0.2 mg $H_3BO_3$ per 1 kg soil), and Mo (as 0.02 mg $(NH_4)_6 Mo_7O_{24}$ per kg soil). The nutrient composition was chosen based on a preliminary experiment with the same soil and the same genotpye of rose plants. Soil samples were air-dried, passed through a 2 mm sieve, and filled into pots at a bulk density of 1.38 $g \cdot cm^{-3}$.

A total of 6 groups of rose plants were treated with different concentrations of MKP (0.0 as a water-only control, 1.0, 2.0, 3.0, 4.0 and 5.0 $g \cdot L^{-1}$) over 2 consecutive flowering cycles. The experiment was set up in a randomized block design with eight replicates in each treatment group. The selected plants were similar in vigour and size. The plants were pruned to a similar height and moved into a greenhouse for the winter to protect them from low temperatures and strong winds. The heated greenhouse maintained temperatures at 15–27 °C during the day and 12–19 °C at night. During the first flowering cycle, MKP fertilisation was initiated when shoot growth started (5 January 2020). Fertilisation was repeated three times at ten-day intervals until flowers started to swell (4 February 2020). The MKP was applied using a combination of foliar spray and soil irrigation. The plants received 50, 50, 100 and 100 mL foliar spray, respectively, combined with 200 mL soil irrigation each time. We used commercial MKP containing 52% $P_2O_5$ and 34% $K_2O$ (23% P and 28% K). In all experiments, foliar application of MKP was performed with a 1 L compression sprayer by wetting the upper and lower surfaces of all leaves. Plants were sprayed until the solution ran off the leaves. Foliar application was performed at sunset, and a non-ionic surfactant (Pulse, Silwet L-77, OSi Specialties Inc., Danbury, CT, USA) was used at a concentration of 0.1% (*v/v*). The basal soil fertilisation was done with a complex fertiliser (15:20:10, N:$P_2O_5$:$K_2O$) at a rate of 1.2 $g \, kg^{-1}$ soil 10 days before the beginning of the treatment for both flowering cycles. Water was added according to the needs of the plants (0.5–2.0 L). Four plants from each group were harvested in the flowering stage on 15 February 2020. The remaining plants were moved outside as the weather became warmer and suitable for rose growth. During the second flowering cycle, rose plants were treated similarly to the first flowering cycle. Different MKP treatments were initiated at the shoot growth stage (15 May 2020) and continued until the flower swelling stage (14 June 2020) at ten-day intervals. The remaining four plants from each group were harvested in the flowering stage on 25 June 2020.

## 2.2. Growth and Flower Production

Plants were harvested in the flowering stage ten days after the fourth MKP application. Flower diameter was measured using a digital vernier calliper. The number of flowers was recorded per plant. Flower dry weight was calculated after samples had been dried. The chlorophyll content of the youngest fully developed leaves was read using a chlorophyll meter (SPAD-502, Minolta, Osaka, Japan).

Plants were separated into shoots and roots. The soil of each pot was sieved through a 2 mm mesh to collect the roots, which were kept in an icebox, transported to the lab, and rinsed with running deionised water. The shoots were divided into new leaves, new stems and flowers, and oven dried at 105 °C for 30 min and then at 70 °C for 3 days to constant weight to determine dry weights and elemental concentration.

## 2.3. Measurements of Root Growth

All roots were scanned by a scanner (Epson Perfection V700 Photo, Epson America, Inc., Long Beach, CA, USA) at a resolution of 400 dpi. Root images were analysed using WinRhizo Pro 2009b software (Regent Instruments Inc., Québec, QC, Canada) to calculate root length and surface area [15]. We classified root branch order according to Pregitzer's method, where the most distal roots were classified as first-order laterals [19]. First-order lateral root density was determined by counting the branches on the representative

axils [20]. The total length of fine roots (diameter < 0.5 mm) was calculated as an index of uptake capacity [21].

### 2.4. Measurement of P and K Accumulation

Dried plant samples were digested in a mixture of 5 mL concentrated $H_2SO_4$ and 5 mL $H_2O_2$ (30%, *v/v*, qualified as chemically pure) [15]. In the digests, we measured P using the molybdo-vanadophosphate method, and K using the flame spectrophotometry method [22]. We used IPE883 straw (Wageningen University, Wageningen, The Netherlands) as reference material to validate the analyses.

In this study, shoot refers to the above-ground part of the plant, including the new leaves, new stems and flowers. We calculated nutrient accumulation as the total P or K uptake by the shoots.

### 2.5. Statistical Analyses

One-way analysis of variance was performed using SAS 8.1 statistical software (SAS Institute, Cary, NC, USA), and significant differences among means were assessed using Tukey's test at 5% probability ($p < 0.05$). Linear models and empirical quadratic equations were produced in SigmaPlot software (v.10.0, Systat Inc., San Jose, CA, USA) to analyse the relationship between MKP concentration, flower production, root growth and total P and K nutrient accumulation.

## 3. Results

### 3.1. Plant Growth Response to MKP Concentration

MKP fertilisation had a significant influence on plant growth and biomass accumulation during both flowering cycles (Table 1). The shoot dry weight in all treatment groups (1.0 to 5.0 $g \cdot L^{-1}$ MKP) increased by 90–153% during the first flowering cycle and by 37–100% during the second flowering cycle compared to that of the water-only control group. During the first flowering cycle, the highest shoot dry weight was recorded in roses fertilised with 3.0 and 4.0 $g \cdot L^{-1}$ MKP, followed by those fertilised with 2.0, 1.0 and 5.0 $g \cdot L^{-1}$ MKP. The lowest shoot dry weight was recorded in the control group. During the second flowering cycle, the highest shoot dry weight was recorded in roses fertilised with 3.0 $g \cdot L^{-1}$ MKP, which was 11–100% greater than that recorded in the other five treatment groups.

**Table 1.** Effects of MKP concentration on shoot and root biomass and leaf SPAD in rose plants during two consecutive flowering cycles.

| Sampling Time | Parameters | MKP Concentration ($g \cdot L^{-1}$) | | | | | |
|---|---|---|---|---|---|---|---|
| | | 0.0 | 1.0 | 2.0 | 3.0 | 4.0 | 5.0 |
| 1st Flowering | Shoot dry weight ($g \cdot plant^{-1}$) | 7.9 (0.26) d | 16 (0.40) bc | 17 (0.40) 4b | 20 (0.79) a | 19 (0.64) a | 15 (0.42) c |
| | Root dry weight ($g \cdot plant^{-1}$) | 3.6 (0.16) d | 5.3 (0.23) c | 6.8 (0.40) b | 9.9 (0.54) a | 6.4 (0.26) bc | 5.3 (0.53) c |
| | Shoot:root ratio | 2.2 (0.13) b | 3.1 (0.18) a | 2.6 (0.21) ab | 2.1 (0.16) b | 3.1 (0.23) a | 2.9 (0.26) a |
| | Leaf SPAD | 36 (1.7) d | 44 (1.2) bc | 46 (1.8) b | 53 (0.83) a | 51 (0.77) a | 41 (1.1) c |
| 2nd Flowering | Shoot dry weight ($g \cdot plant^{-1}$) | 30 (1.1) d | 46 (2.1) bc | 54 (2.8) ab | 60 (3.4) a | 48 (3.2) bc | 41 (4.6) c |
| | Root dry weight ($g \cdot plant^{-1}$) | 6.7 (0.31) b | 7.8 (0.71) ab | 10.9 (2.2) a | 10.2 (1.2) ab | 8.1 (0.80) ab | 7.2 (0.85) b |
| | Shoot:root ratio | 4.5 (0.36) a | 6.0 (0.45) a | 5.3 (0.97) a | 6.1 (1.0) a | 6.0 (0.30) a | 6.0 (1.2) a |
| | Leaf SPAD | 43 (1.9) c | 50 (0.75) b | 51 (0.69) b | 55 (0.70) a | 56 (0.47) a | 49 (0.83) b |

Each value is the mean of four replicates. Different letters in each row denote significant difference among treatments ($p < 0.05$). MKP, $KH_2PO_4$; 1st Flowering, the first flowering cycle; 2nd Flowering, the second flowering cycle.

MKP fertilisation significantly improved root growth during the first flowering cycle. During the first flowering cycle, the root dry weight of roses fertilised with MKP was 47–175% greater compared to that of roses that only received water. In contrast, there was no significant difference in root dry weight between MKP treatment groups and the control group during the second flowering cycle, except for the 2.0 g·L$^{-1}$ MKP treatment group, which had a 7–63% greater root dry weight compared to other treatment groups. During the first flowering cycle, the 3.0 g·L$^{-1}$ treatment group had a significantly decreased shoot:root ratio compared to the other four MKP treatment groups, indicating a higher relative root biomass in the 3.0 g·L$^{-1}$ MKP treatment group. In contrast, during the second flowering cycle, there was no difference in the shoot:root ratio among the six treatment groups.

During the first flowering cycle, leaf SPAD increased as MKP concentration increased from 0 to 3.0 g·L$^{-1}$, remained constant with 4.0 g·L$^{-1}$ MKP fertilisation, but markedly decreased with 5.0 g·L$^{-1}$ MKP fertilisation (Table 1). During the first flowering cycle, leaf SPAD increased by 15–47% in response to 3.0 g·L$^{-1}$ MKP fertilisation and by 11–42% in response to 4.0 g·L$^{-1}$ MKP fertilisation compared to the control group and other MKP treatment groups. Leaf SPAD showed a similar trend during the second flowering cycle as during the first flowering cycle.

### 3.2. Flower Growth Response to MKP Concentration

Rose flower growth and development were stimulated by 1.0 to 5.0 g·L$^{-1}$ MKP fertilisation (Figure 1). During the first flowering cycle, flower dry weight increased by 176–375%, flower diameter increased by 193–565% and flower numbers per plant increased by 50–83% in roses fertilised with 1.0 to 5.0 g·L$^{-1}$ MKP compared to roses that only received water. The greatest flower dry weight was recorded in roses fertilised with 2.0 and 3.0 g·L$^{-1}$ MKP, followed by those treated with 4.0, 1.0 and 5.0 g·L$^{-1}$ MKP and the control treatments. Flower diameter significantly increased as the concentration of MKP fertiliser increased from 0.0 to 3.0 g·L$^{-1}$ MKP treatment but decreased in response to fertilisation with higher MKP concentrations. The number of flowers per plant significantly increased as the concentration of MKP fertiliser increased from 0 to 2.0 g·L$^{-1}$ and remained constant with the application of higher concentrations of MKP fertiliser. During the second flowering cycle, flower dry weight showed a similar trend as during the first flowering cycle. The highest flower dry weight was found in roses fertilised with 2.0 and 3.0 g·L$^{-1}$ MKP. Flower diameter gradually increased as MKP concentration increased from 0.0 to 3.0 g·L$^{-1}$ and slightly but significantly decreased in response to 4.0 and 5.0 g·L$^{-1}$ MKP. There was no difference in flower diameter between roses fertilised with 5.0 g·L$^{-1}$ MKP and those in the control groups. The greatest number of flowers per plant was recorded in roses that received 3.0 and 4.0 g·L$^{-1}$ MKP, followed by those that received 2.0, 5.0, and 1.0 g·L$^{-1}$ MKP and water only.

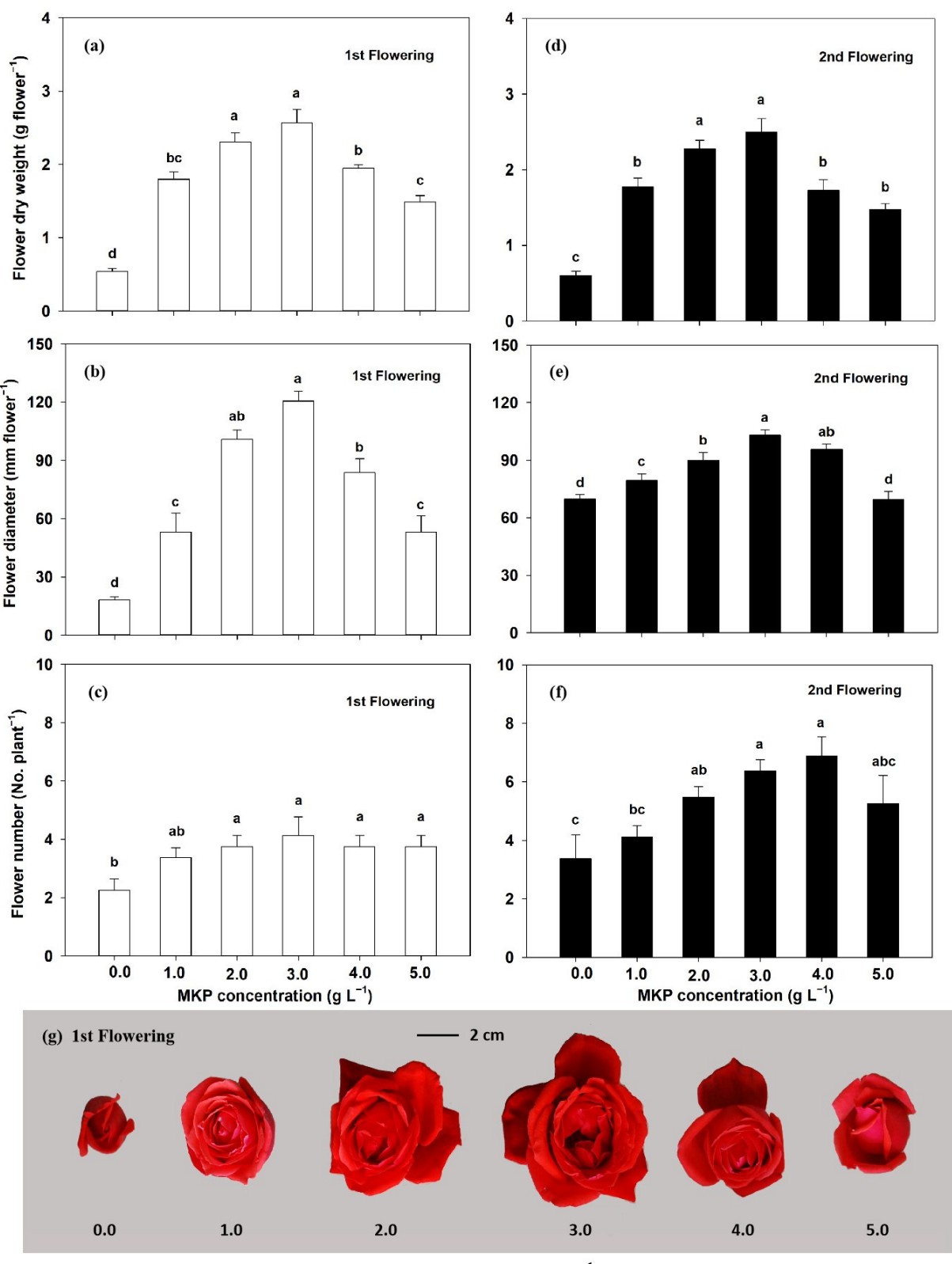

**Figure 1.** Effects of MKP concentration on flower dry weight (**a**,**d**), flower diameter (**b**,**e**) and flower number (**c**,**f**) of rose plants during the both flowering cycles, as well as flower feature (**g**) during the first flowering cycle. Each value is the mean of four replicates (+SE). Significant differences ($p < 0.05$) among the six MKP treatments were denoted by different lowercase letters. MKP, $KH_2PO_4$; 1st Flowering, the first flowering cycle; 2nd Flowering, the second flowering cycle.

### 3.3. Root Growth Response to MKP Concentration

During the first flowering cycle, total root length gradually increased as the concentration of MKP fertiliser increased from 0.0 to 3.0 g·L$^{-1}$ MKP, but fertilisation with higher MKP concentrations lead to a significant decrease in total root length (Figure 2). Total root surface area and total fine root length exhibited a similar trend as total root length. Total root length increased by 51–185%, total root surface area by 74–202% and total fine root length by 50–194% in response to 3.0 g·L$^{-1}$ MKP fertilisation, compared to the other five treatment groups. First-order lateral root density was 66–239% greater in roses that received 1.0 to 3.0 g·L$^{-1}$ MKP fertilisation compared to those in the control group and treatment groups that received higher MKP concentrations.

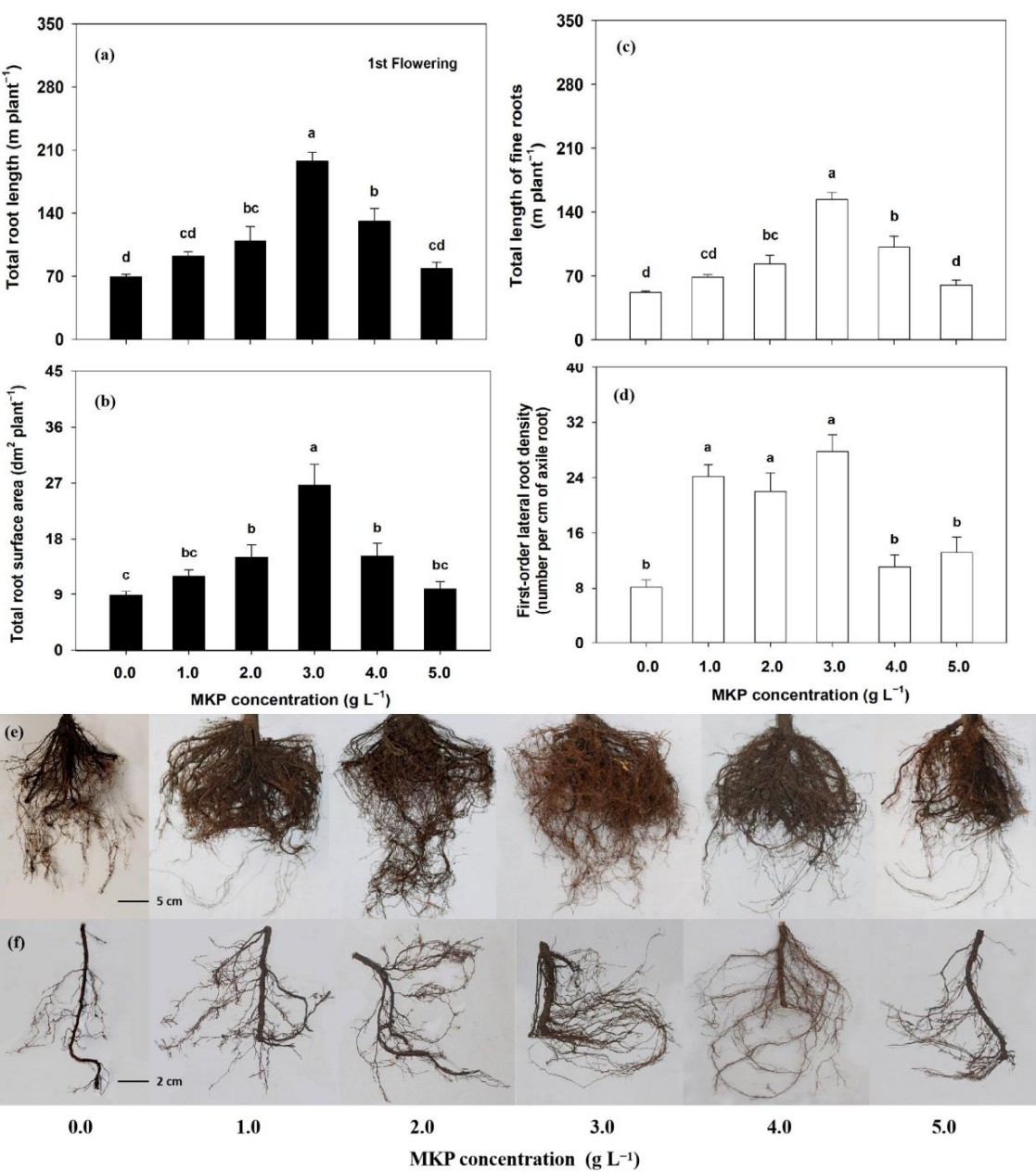

**Figure 2.** Total root length (**a**), total root surface area (**b**), total fine root length (**c**), first-order lateral root density (**d**), and characteristics of the whole roots and lateral roots (**e**,**f**) of rose plants in response to different concentrations of MKP during the first flowering cycle. Each value is the mean of four replicates (+SE). Different letters denote significant differences ($p < 0.05$) among the six MKP treatments. MKP, $KH_2PO_4$; 1st Flowering, the first flowering cycle.

MKP fertilization also had a significant effect on root growth during the second flowering cycle (Figure 3). The highest total root length was recorded in roses fertilised with 2.0 g·L$^{-1}$ MKP, followed by those fertilised with 3.0 g·L$^{-1}$. No significant differences in total root length were found among the 1.0, 4.0, and 5.0 g·L$^{-1}$ MKP and water-only control treatment groups. Total root length was 57–109% greater in the 2.0 g·L$^{-1}$ MKP treatment group and 24–64% greater in the 3.0 g·L$^{-1}$ MKP treatment groups. Total root surface area was 54–104% greater in the 2.0 g·L$^{-1}$ MKP treatment group and 12–49% greater in the 3.0 g·L$^{-1}$ MKP treatment group compared to other treatment groups.

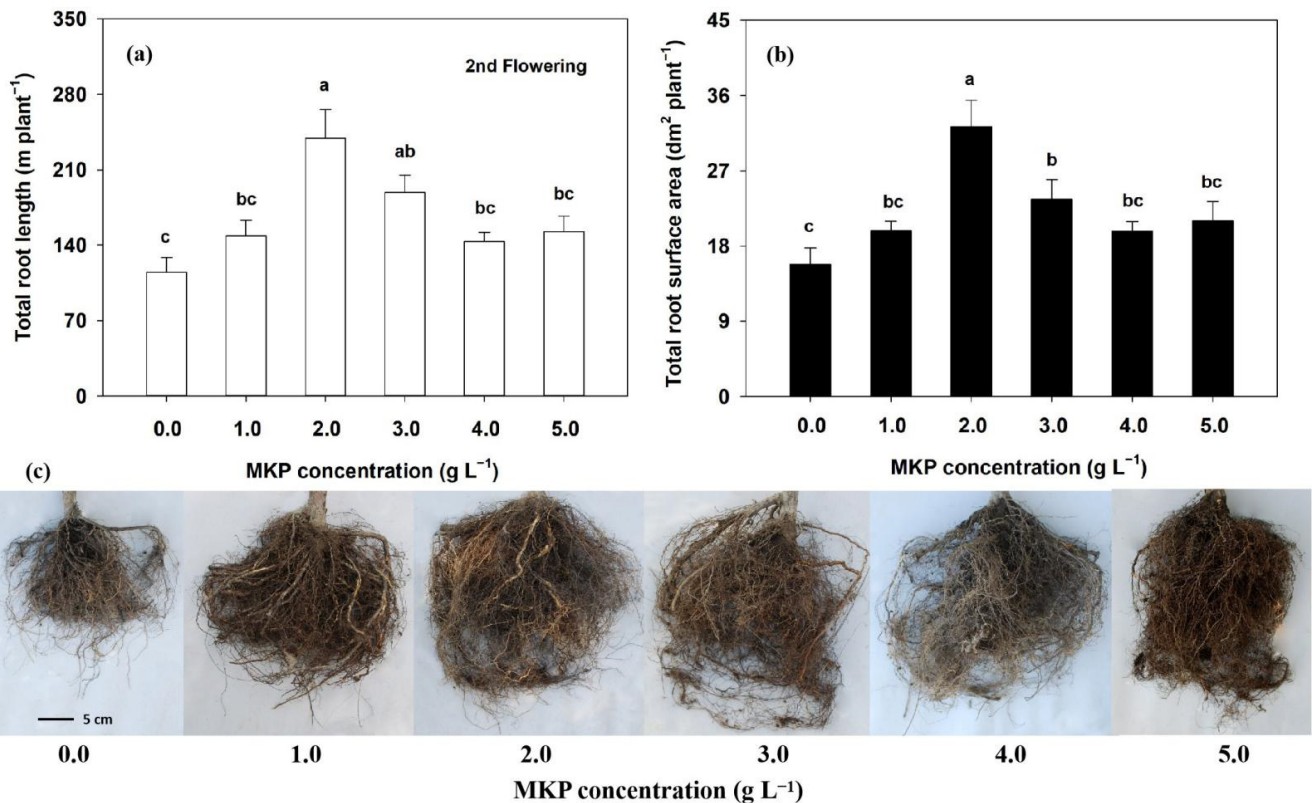

**Figure 3.** Total root length (**a**), total root surface area (**b**) and characteristics of the whole roots (**c**) of rose plants in response to different concentrations of MKP during the second flowering cycle. Each value is the mean of four replicates (+SE). Different letters denote significant difference ($p < 0.05$) among the six MKP treatments. MKP, KH$_2$PO$_4$; 2nd Flowering, the second flowering cycle.

### 3.4. Plant P and K Accumulation Response to MKP Concentration

MKP fertilisation had a positive influence on plant P and K accumulation during both flowering cycles (Figure 4). During the first flowering cycle, total P and K accumulation in shoots increased by 27–68% and 64–101%, respectively, in response to 1.0 to 5.0 g·L$^{-1}$ MKP fertilisation compared to the water-only control group. P and K accumulation in flowers were 49–89% and 171–248% greater, respectively, in response to 1.0–5.0 g·L$^{-1}$ MKP fertilisation compared to the control group. P and K accumulation in new leaves and stems showed a similar trend to that in flowers.

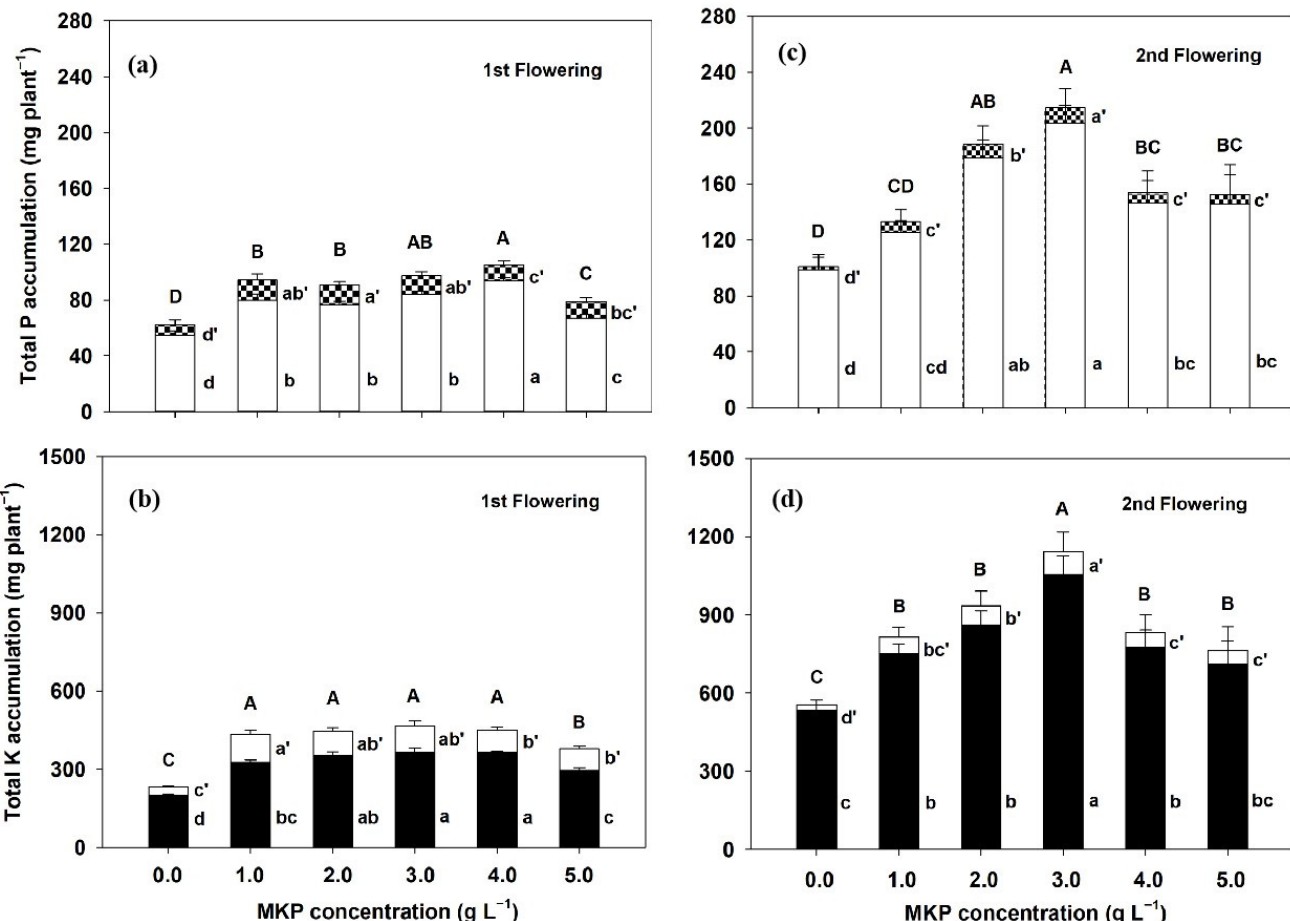

**Figure 4.** Effects of MKP concentration on total P (**a,c**) and K (**b,d**) accumulation of rose plants during the first and second flowering cycle. Significant differences ($p < 0.05$) of P and K accumulation in shoots and flowers among the six treatments were denoted by different lowercase letters without (for shoots) and with apostrophes (for flowers) and by different capital letters for the above-ground parts of the plants. MKP, $KH_2PO_4$; 1st Flowering, the first flowering cycle; 2nd Flowering, the second flowering cycle.

During the second flowering cycle, total P and K accumulation gradually increased as MKP concentration increased from 0.0–3.0 g·L$^{-1}$ MKP, but decreased in response to 4.0 and 5.0 g·L$^{-1}$ MKP fertilisation. Total P and K accumulation increased by 14–113% and 22–106%, respectively, compared to other treatment groups in response to 3.0 g·L$^{-1}$ MKP fertilisation. P and K accumulation in flowers, new leaves and new stems showed a similar trend as the total nutrient accumulation.

MKP fertilisation had a positive influence on plant N accumulation. Plant N accumulation exhibited a similar trend as P and K accumulation during both flowering cycles. During the first flowering cycle, total N accumulation gradually increased as MKP concentration increased from 0.0–4.0 g·L$^{-1}$ MKP, but decreased in response to 5.0 g·L$^{-1}$ MKP fertilisation. During the second flowering cycle, total N accumulation gradually increased as the concentration of MKP fertiliser increased from 0.0–3.0 g·L$^{-1}$ MKP, but decreased with higher MKP concentration (data not shown).

*3.5. Relationship between Flower Characteristics, Nutrient Accumulation, Root Length and MKP Concentrations*

Both flower characteristics (diameter and dry weight) and total nutrient accumulation (P and K) were significantly correlated with MKP concentrations ($p \leq 0.01$) during the first flowering cycle (Figure 5). According to empirical quadratic equations, 2.8, 3.0, 2.7, and

$2.9 \text{ g·L}^{-1}$ MKP fertilisation would produce the optimal flower diameter, flower dry weight, and total P and K accumulation, respectively. A similar trend was observed during the second flowering cycle, where optimal results would be produced by $2.6–2.9 \text{ g·L}^{-1}$ MKP fertilisation (data not shown). Additionally, total root length significantly correlated with flower diameter ($R^2 = 0.63$, $p \leq 0.01$), flower dry weight ($R^2 = 0.70$, $p \leq 0.01$) and total P ($R^2 = 0.76$, $p \leq 0.01$) and K accumulation ($R^2 = 0.67$, $p \leq 0.01$) during the first flowering cycle (data not shown). Similar trends were observed during the second flowering cycle.

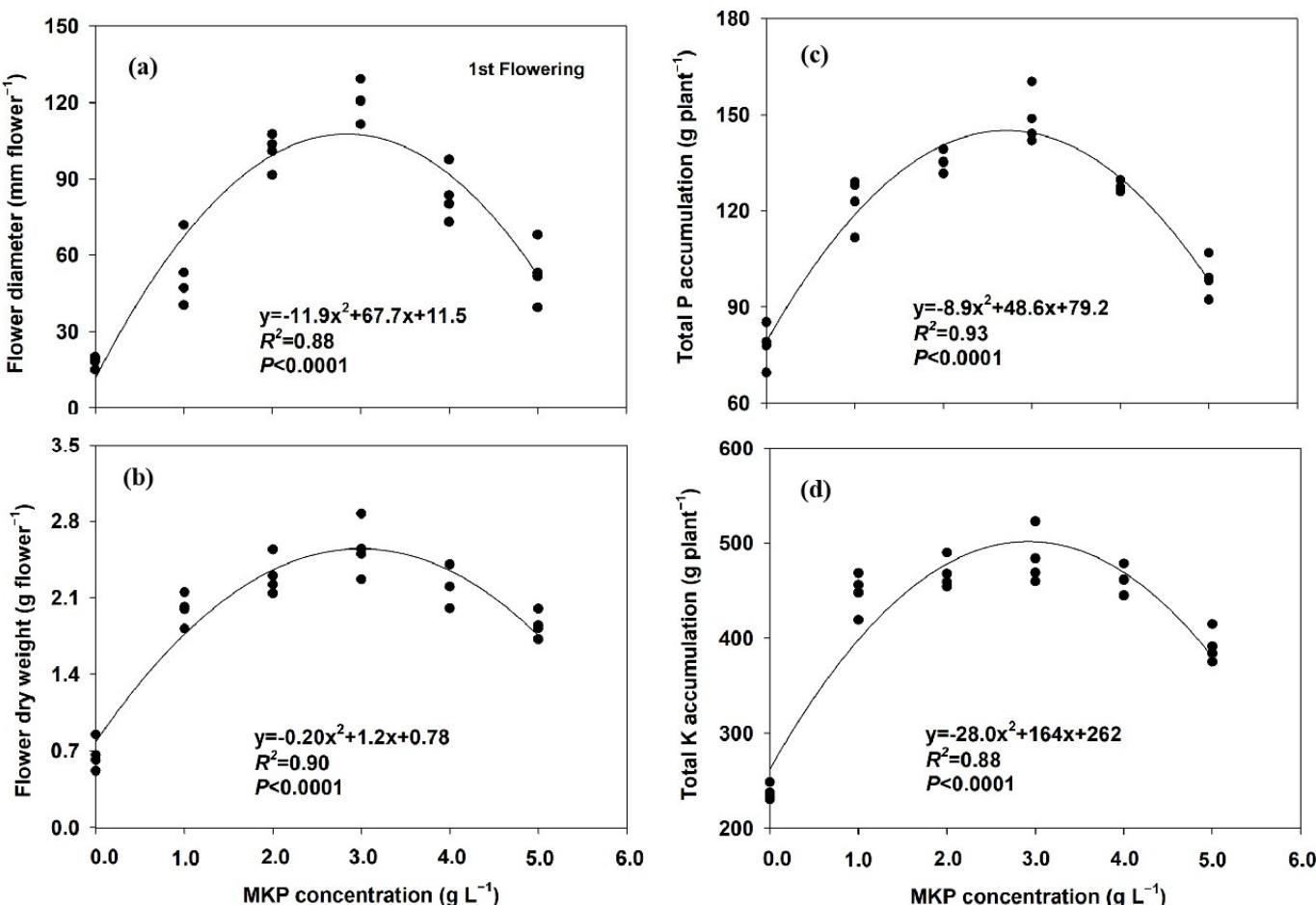

**Figure 5.** Relationships of flower diameter and dry weight (**a**,**b**) and total P and K nutrient accumulation (**c**,**d**) with different MKP concentrations during the first flowering cycle. MKP, $KH_2PO_4$; 1st Flowering, the first flowering cycle.

## 4. Discussion

### 4.1. Effects of MKP Concentration on Plant Growth and Flower Production

MKP is a fully water-soluble composite and an excellent source of P and K widely used in horticulture [10,18]. Several studies have reported a positive effect of MKP fertilisation on plant growth and the quality and quantity of flowers and fruit [9,10]. In the present study, the results clearly demonstrated that fertilisation with the optimal concentration of MKP significantly improved rose growth and flower production. It has been recognised that rose flower production and nutrient efficiency can be improved by adopting efficient management strategies, including optimal concentrations and application methods.

In the present study, shoot dry weight gradually increased as MKP concentration increased from $0.0–3.0 \text{ g·L}^{-1}$, but significantly decreased with 4.0 or $5.0 \text{ g·L}^{-1}$ MKP fertilisation during both flowering cycles. In *Rosa indica*, a $3.4 \text{ g·L}^{-1}$ MKP foliar spray improved plant growth and reduced disease incidence of *Sphaerotheca pannosa* [23]. The application of MKP concentrations that were either lower or higher than optimal values had

adverse effects on growth and flower quality [16,24]. The stimulatory effect of MKP on vegetative growth in all the MKP-treated roses was accompanied by increased chlorophyll content in the leaves (Table 1) and a greater leaf area compared to roses that received water alone [25]. Increased chlorophyll concentration in the leaves of MKP-fertilised plants could cause increased shoot biomass because increased chlorophyll concentration may hasten photosynthetic activities and other physiological processes related to plant growth and development.

In general, MPK fertilisation had a positive effect on flower diameter and dry weight during both flowering cycles. Fertilisation with 2.0 or 3.0 g·L$^{-1}$ MKP produced the greatest flower diameter and dry weight, but flower diameter and dry weight were reduced in response to fertilisation with excessive (4.0 and 5.0 g·L$^{-1}$) or insufficient (0.0 and 1.0 g·L$^{-1}$) concentrations of MKP (Figure 1). These results are supported by findings that flowering intensity was highly dependent on the concentration of N, P and K in the irrigation solution [26]. Another study reported that foliar application of MKP stimulated flowering in Litchi trees, with a significantly increased number of male, female and total flowers per panicle [9]. In the present study, a significant quadratic relationship between flower diameter and dry weight and MKP concentration indicated that the application of 2.7–3.0 g·L$^{-1}$ MKP would optimise flowering, growth and development.

This study showed that the application of the optimal concentration of MKP had a positive effect on plant growth and flower production compared to treatment with water only. Possible explanations for the positive effect of MKP fertilisation include: (1) a combination of foliar and basal soil MKP treatments could provide a sufficient amount of P and K nutrients for plant growth, which may be essential for optimising yield [7]; (2) foliar application of MKP provides nutrients to plants through mist in a straight line onto foliage and flora [27], and soil application of MKP may improve root growth and nutrient uptake; (3) the time of fertilisation (from shoot elongation to flower bud swelling stage) coincides with a stage of critical nutrient requirements of *R. damascena* [28], which may contribute to the nutrient demand of and replenish the nutrient supply in the root tissues [29]; (4) the optimal concentration of MKP may increase the effective absorbance of P and K nutrients and improve physiological activities essential for flower growth and development [30]; (5) increased N accumulation was observed with the optimal MKP concentration (data not shown), which would have been associated with plant growth and flower production [26]. Studies have showed that availability of N, P and K influenced the quantity and quality, such as intensity and longevity, of cut and potted flowers [26,31]. Nitrogen is essential for protein biosynthesis, and developing inflorescences have been shown to be powerful sinks for N and water-soluble proteins [5]. Furthermore, the possible mechanisms integrating N-P or N-K interactive regulation pathways were reported, which suggested that the coordinated acquisition of mineral nutrients is essential to maintain optimal plant growth and achieve maximal yield [32,33].

### 4.2. Effects of MKP Concentration on Root Growth and Nutrient Uptake

Plants show plasticity in shoot and root growth to adapt to heterogeneous nutrient supply and changing environments [34]. Roots exhibit a high degree of physiological and morphological plasticity to capture nutrients in the soil. A previous study showed that P levels had an important effect on root growth and P accumulation in *R. multiflora* [35]. Similarly, root growth in R. multiflora rootstocks showed high plasticity in response to different concentrations of MKP in the present study. Furthermore, morphological traits such as total root length and total root surface area were significantly correlated with MKP concentrations ($p \leq 0.01$). These results suggested that enhanced root growth in response to optimal MKP concentrations could have contributed to increased nutrient uptake and accumulation.

In the present study, total root length and root surface area were significantly correlated with total P and K accumulation ($p < 0.05$), which was associated with increased total fine root length. Additionally, fine roots accounted for 75% of the total length of the root

system in the present study (Figure 2). It is well known that fine roots are important organs for nutrient and water absorption from the soil. Therefore, fertilisation with the optimal concentration of MKP may increase nutrient absorption and transportation to shoots, leading to increased N, P and K accumulation. Compared to the 3.0 $g \cdot L^{-1}$ MKP treatment group, root length and surface area were reduced in the 1.0 and 2.0 $g \cdot L^{-1}$ MKP treatment groups, even if the first-order lateral root density was significantly increased (Figure 2). Decreased shoot biomass may be responsible for inhibited root growth following 1.0 or 2.0 $g \cdot L^{-1}$ MKP treatment (Table 1). Root growth is highly dependent on the supply of carbohydrates from the leaves; thus, root growth is likely to be decreased when the supply of carbohydrates is insufficient [36–38].

In the present study, treatment with 1.0–5.0 $g \cdot L^{-1}$ MKP improved total P and K accumulation in shoots. Furthermore, P and K accumulation were significantly correlated with MKP concentration ($p \leq 0.01$), and could reach optimal values following fertilisation with 2.7–2.9 $g \cdot L^{-1}$ MKP according to empirical quadratic equations of both flowering cycles. Similar studies showed that MKP soil fertilisation improved plant growth, nutrient utilisation, and yield [27,39]. In contrast, spraying mango or olive trees with 20.0 $g \cdot L^{-1}$ or 30.0 $g \cdot L^{-1}$ MKP improved vegetative parameters and P and K contents in leaves [25,40]. A foliar spray of 1.7 $g \cdot L^{-1}$ di-potassium hydrogen orthophosphate or 2.0 $g \cdot L^{-1}$ MKP improved plant growth and nutrient utilization in eggplant and tall fescue under adverse conditions [17,18]. The contradictory results may be due to differences in species, genotypes, management practices, or the growth stage [8,41]. Future studies could focus on optimising MKP fertilisation practices and understanding the underlying mechanisms based on plant genotypes, root growth, nutrient availability, etc. to increase yield and improve nutrient use efficiency.

## 5. Conclusions

Various concentrations (1.0–5.0 $g \cdot L^{-1}$) of MKP fertiliser administered with a combination of foliar and soil applications showed different effects on rose growth, flower production and nutrient uptake. Rose shoot biomass, leaf SPAD, flower production (diameter and biomass), root characteristics (root length and surface area), as well as P and K accumulation significantly increased as the concentration of MKP increased from 0.0 to 2.0 or 3.0 $g \cdot L^{-1}$ but decreased in response to 4.0 or 5.0 $g \cdot L^{-1}$ MKP fertilisation during both flowering cycles. According to quadratic equations, 2.6–3.0 $g \cdot L^{-1}$ is the optimal concentration of MKP for flower diameter, dry weight and total P and K accumulation in roses. Furthermore, total root length was significantly correlated with flower diameter, dry weight and total P and K accumulation, indicating that fertilisation with the optimal concentration of MKP can increase rose growth, flower production and nutrient uptake through enhanced root growth.

**Author Contributions:** Q.M., X.W. and M.L. designed the study; Q.M., X.W., W.Y. and M.L. were in charge of the data collection; Q.M. and H.T. analyzed the data and drafted the paper; Q.M., X.W. and M.L. reviewed and edited the final version of paper; Q.M. acquired the funding. All authors have read and agreed to the published version of the manuscript.

**Funding:** This research was funded by the Chinese National Natural Science Foundation (No. 31700559).

**Institutional Review Board Statement:** Not applicable. This study is not involving humans or animals.

**Informed Consent Statement:** Not applicable.

**Data Availability Statement:** Data are contained within the article.

**Acknowledgments:** We thank Xiaoqiang Jiao, China Agricultural University.

**Conflicts of Interest:** The authors declare no conflict of interest. The founding sponsors had no role in the design of the study; in the collection, analyses, or interpretation of data; in the writing of the manuscript, and in the decision to publish the results.

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
