# Peer review of "The Optimal Concentration of KH2PO4 Enhances Nutrient Uptake and Flower Production in Rose Plants via Enhanced Root Growth"

_agriculture, doi:10.3390/agriculture11121210_

Round 1

Reviewer 1 Report

Dear Author

The research reported in this paper is of significance to improve the growth and production of Rose. However, I find the material and method section a bit confusing and needs more clarity. Among the nutrients only potassium and phosphorus are studied, I think adding nitrogen status would also add to the result. Both potassium and phosphorus influence the status of nitrogen, which eventually affects the quality of flowers.

Nutrients are known to impact flower longevity/withering and therefore, if you could add text related to potassium, nitrogen and phosphorus in the discussion.

Author Response

  1. The research reported in this paper is of significance to improve the growth and production of Rose. However, I find the material and method section a bit confusing and needs more clarity.

Response: We are very grateful for the comment. We have added relative references to the all methods in line 174-181. 

  1. Among the nutrients only potassium and phosphorus are studied, I think adding nitrogen status would also add to the result. Both potassium and phosphorus influence the status of nitrogen, which eventually affects the quality of flowers.
    Response: Thank you very much. MKP fertilisation had a positive influence on plant N accumulationin the present study. Plant N accumulation exhibited a similar trend as P and K accumulation during both flowering cycles. During the first flowering cycle, total N accumulation gradually increased as MKP concentration increased from 0.0 to 4.0 g.L-1MKP, but decreased in response to 5.0 g.L1 MKP fertilisation. During the second flowering cycle, total N accumulation gradually increased as the concentration of MKP fertiliser increased from 0.0 to 3.0 g.L-1 MKP, but decreased with higher MKP concentration (data not shown).We have added relevant information in line 303-309.

  1. Nutrients are known to impact flower longevity/withering and therefore,if you could add text related to potassium, nitrogen and phosphorus in the discussion.
    Response: Good comments.Increased N accumulation was observed with the optimal MKP concentration (data not shown), which would have been associated with plant growth and flower production [26]. Studies have showed that availability of N, P and K influenced the quantity and quality such as intensity and longevity of cut and potted flowers [26,31]. Nitrogen is essential for protein biosynthesis, and that developing inflorescences have been shown to be powerful sinks for N and water-soluble proteins [5]. Furthermore, the possible mechanisms integrating N-P or N-K interactive regulation pathways were reported, which suggested the coordinated acquisition of mineral nutrients is essential to maintain optimal plant growth and achieve maximal yield [32,33].We have added relevant information in line 372-381.

[31] Ahmad, I.; Dole, J.M.; Nelson, P. Nitrogen application rate, leaf position and age affect leaf nutrient status of five specialty cut flowers. Sci. Hortic. 2012. 142, 14-22.

[32] Hu, B.; Chu, C. Nitrogen–phosphorus interplay: old story with molecular tale. New Phytol. 2020. 225, 1455-1460.

[33] Drechsler, N.; Zheng, Y.; Bohner, A.; Nobmann, B.; Wirén, N.; Kunze, R.; Rausch, C. Nitrate-dependent control of shoot K homeostasis by NPF7.3/NRT1.5 and SKOR in Arabidopsis. Plant Physiol. 2015, 169, 2832-2847.

We have made a thorough revision in the revised manuscript. We would like to publish it in the journal of Forests. Future studies could focus on optimising MKP fertilisation practices and understanding the underlying mechanisms based on plant genotypes, root growth, nutrient availability, etc. to increase yield and improve nutrient use efficiency.

Reviewer 2 Report

  1. Please correct editing errors, g. KH2PO4 (L. 2, L. 63), mg.kg−1 (L. 109), CaCl2, K2SO4 (L. 112), MgSO4 (L. 113), etc.
  2. Is it correct? : 25 June 2019 (L. 143)
  3. On what basis were the doses of mineral fertilization determined?
  4. Please add references to the all methods used
  5. All abbreviations and acronyms used in tables and figures should be defined in the table notes or figure captions.

Author Response

  1. Please correct editing errors, g. KH2PO4 (L. 2, L. 63), mg.kg−1 (L. 109), CaCl2, K2SO4 (L. 112), MgSO4 (L. 113), etc.
    Response: Thank you very much. We have revised all the editing errors, g. KH2PO4(L.2, L.75), mg.kg−1(L.123-124), CaCl2, K2SO4 (L.126-127), MgSO4 (L.127) and all the concentrations of MKP (0.0 as a water-only control, 1.0, 2.0, 3.0, 4.0 and 5.0 g.L-1)  etc.
  2. Is it correct? : 25 June 2019 (L. 143).
    Response: We are grateful for the comment. We changed the error, and it has been changed in line 159as follows: 25 June 2020.
  3. On what basis were the doses of mineral fertilization determined?
    Response: Good comments. We have added relevant information in line 130-131as follows:The nutrient composition was chosen based on a preliminary experiment with the same soil and the same genotpye of rose plants.
  4. On Please add references to the all methods used.
    Response: Thanks. We have added relevant information in line 174-181.
  5. All abbreviations and acronyms used in tables and figures should be defined in the table notes or figure captions.
    Response: We really appreciated the suggestion. All abbreviations and acronyms used in tables and figures havebe defined in the notesof table 1 and figures (Figs. 1, 2, 3,4 and 5) .